

# Northern North Atlantic climate variability controls on ocean carbon sinks in EC-Earth3-CC

Anna Pedersen[1,2], Carolin R. Löscher[1], Steffen M. Olsen[2]

[1]Nordcee, University of Southern Denmark, Odense, 5230, Denmark

[2]Danish Meteorological Institute, National Centre for Climate Research, Copenhagen, 2100, Denmark

*Correspondence to*: Anna Pedersen (annapedersen@biology.sdu.dk)

## Abstract

The northern North Atlantic is an important net sink of atmospheric $CO_2$, though air-sea $CO_2$ fluxes exhibit substantial variability across different timescales. The underlying drivers of this variability remain poorly understood across both

temporal and regional scales. Here, we investigate interannual to decadal $CO_2$ flux variability in the northern North Atlantic using historical simulations from the EC-Earth3-CC model. We assess the role of key dynamical and physical processes in shaping $CO_2$ flux variability across five regions: the Nordic Seas, eastern Nordic Seas, the eastern and western subpolar North Atlantic, and the full North Atlantic. Our analysis reveals that physical parameters—including sea ice concentration (SIC), sea surface temperature (SST), sea surface salinity (SSS), and wind stress—along with dynamical processes related to

ocean mixing and circulation, play a central role in regulating $CO_2$ flux variability. Using regression analysis, we demonstrate that these drivers exert regionally and temporally varying influences, with our models achieving high $R^2$ values indicating a strong degree of explanation for $CO_2$ flux variability. The regression models capture interannual variability more effectively than decadal variability, highlighting the dominant role of short-term fluctuations in shaping $CO_2$ flux dynamics. Overall, our results demonstrate that the predictors of $CO_2$ flux variability are both spatially and temporally dependent. We

find that $CO_2$ flux variability cannot be fully explained by simple linear correlations with individual predictors but instead arises from complex interactions among multiple physical and dynamical processes. Notably, $CO_2$ flux variability is particularly sensitive to changes in certain predictors, such as wind stress, consistent with expectations based on the gas transfer equation used to compute air-sea $CO_2$ fluxes.



## 1. Introduction

The northern North Atlantic is a net sink of atmospheric carbon dioxide ($CO_2$; Fig. 1a; Gruber et al., 2002; Takahashi et al., 2009, Yu et al., 2019). This sink is primarily driven by the combined effects of the solubility pump and the biological carbon pump. The solubility pump enhances $CO_2$ uptake through two key mechanisms: (1) the cooling of northward-flowing warm waters, which increases $CO_2$ solubility at the surface, and (2) the formation of dense, cold deep water at high latitudes, which transports $CO_2$-enriched surface waters into the ocean interior as part of the Atlantic thermohaline circulation (Volk et al., 1985). The biological carbon pump further contributes by fixating carbon through photosynthesis and primary production (PP), exporting organic carbon to the deep ocean, and maintaining a net sink of atmospheric $CO_2$ in the region (Sigman and Boyle, 2000; Boyd and Trull, 2007; Sanders et al., 2014). Currently, the North Atlantic absorbs a significant fraction of anthropogenic $CO_2$ emissions, accounting for approximately one-quarter to one-third of global anthropogenic carbon uptake (Khatiwala et al., 2009; Sabine et al., 2004; Breeden and McKinley, 2016; Le Quéré et al., 2018; Gruber et al., 2019). In contrast, the global ocean uptake is around 25% of annual emissions due to outgassing in some ocean regions (Fig. 1; Landschützer et al., 2016; Mikaloff Fletcher et al., 2006; Khatiwala et al., 2013; Le Quéré et al., 2015; Friedlingstein et al., 2023; Gruber et al., 2023; Terhaar et al., 2022).

Beyond these large-scale mechanisms, the strength of the North Atlantic $CO_2$ sink varies seasonally and interannually, influenced by ocean-atmosphere interactions. Biological productivity peaks in spring and summer, enhancing $CO_2$ drawdown, while wintertime cooling promotes solubility-driven uptake (Ardyna and Arrigo 2020). These processes are modulated by climate variability, particularly the North Atlantic Oscillation (NAO) and the Atlantic Multidecadal Oscillation (AMO; Leseurre et al., 2020). Additionally, physical processes such as warming and cooling cycles, deep convection, and changes in ocean circulation and water masses in the subpolar gyre (SPG) play a crucial role in regulating air-sea $CO_2$ fluxes. These physical drivers are closely linked to NAO and AMO variability, further influencing regional $CO_2$ uptake dynamics (Chafik et al., 2019; Desbruyëres et al., 2015). Breeden and McKinley (2016) show that a North Atlantic basin-average sea surface temperature (SST) is associated with the leading mode of surface ocean $pCO_2$ variability based on an analysis of a regional model. The SST signal is affected by an upward trend due to greenhouse gas emissions and a signal of internal variability due to the AMO (Kerr, 2000). Furthermore, they establish that physical (in contrast to the chemical) variability is the dominant driver of variability in the North Atlantic surface ocean carbon cycle. During a positive phase of the AMO an increase in sea (surface) temperature increases $pCO_2$ due to the reduction in solubility (an increase in the fugacity of $CO_2$, $fCO_2$) resulting in a reduced flux of $CO_2$ into the ocean (also seen in Breeden and McKinley (2016)). Studies show that on a decadal scale the AMO and $fCO_2$ variability in the SPG correlates (Breeden and McKinley, 2016; Landschutzer et al., 2019; Leseurre et al., 2020): as the AMO enters a positive (warm) phase, reduced mixing (a shallowing of the mixed layer depth) results in a reduced supply of dissolved inorganic carbon (DIC) from the subsurface layers, a process that dominates the effect of warming on the solubility and $fCO_2$. The surface ocean's ability to absorb anthropogenic carbon is primarily regulated by carbonate chemistry, particularly alkalinity (Broecker et al., 1979; Terhaar et al., 2022).



While increasing anthropogenic carbon generally enhances ocean carbon uptake, this relationship is moderated by rising
global temperatures, which reduce solubility, and further influenced by decadal variability and long-term trends. However,
the underlying drivers of these trends remain poorly understood (Terhaar, 2024).

The subpolar North Atlantic has experienced significant interannual to decadal variability in ventilation depth
during the industrial period (Polyakov 2005; Holliday et al., 2020; Zou et al., 2023, Thomas and Zhang 2022). Recent
studies highlight pronounced ocean variability in the Subpolar Gyre region, including the formation of cold and fresh
anomalies. Holliday et al. (2020) documented a substantial freshening event in the eastern subpolar North Atlantic, the most
significant in 120 years, attributed to wind-driven circulation changes that redistributed freshwater within the region.
Similarly, Zou et al. (2023) identified two sources of deep decadal variability in the central Labrador Sea, furthering our
understanding of long-term property anomalies in the western subpolar North Atlantic. These findings underscore the SPG's
dynamic nature and its sensitivity to both atmospheric and oceanic forcing. The SPG, a counterclockwise gyre system, plays
a crucial role in modulating ocean circulation by introducing saline Atlantic Water into the northern North Atlantic. The
North Atlantic Current follows the southern and eastern boundaries of the SPG, transporting warm, saline water from the
Gulf Stream into the northern North Atlantic and eventually the Arctic (Holliday et al., 2020; Daniault et al., 2016; Hátún et
al., 2017). At the same time, freshwater from the Arctic is supplied to the SPG, contributing to temporal variability in ocean
properties. These processes are key regulators of the AMOC (Holliday et al., 2020; Hátún et al., 2005, 2017). The strength of
the SPG circulation strongly influences both physical and biological processes in the northern North Atlantic. A weak
(strong) SPG leads to a shallowing (deepening) of sea surface height (SSH) and, consequently, a shallowing (deepening) of
the mixed layer depth (MLD). Additionally, a weak SPG contracts westward, allowing nutrient-poor subtropical waters to
penetrate northward, warming the gyre and reducing primary production. In contrast, a strong SPG expands eastward,
accumulates Arctic fresh and nutrient-rich water within the gyre system, increasing PP and soliubility, and limits the
intrusion of warm, saline Atlantic Water into the region (Häkkinen et al., 2013; Hátún et al., 2005, 2017; Foukal et al., 2017).
Importantly, variations in SSS serve as an indicator of these gyre dynamics and ocean mixing processes, making it a valuable
indirect predictor of $CO_2$ flux variability. Changes in salinity reflect shifts in water mass distribution, stratification, and
ventilation, all of which influence air-sea gas exchange. Therefore, monitoring SSS variability can provide critical insights
into the mechanisms driving $CO_2$ flux variability in the North Atlantic (Thomas and Zhang, 2022).

Understanding the vulnerability of the ocean carbon sink to future climatic changes is the motivator for our
analysis. Expected future decline in the strenght of the Atlantic Meriodional Ocean Circulaiton (AMOC; Fox-Kemper et al.,
2023) will likely result in a declining ocean $CO_2$ uptake due to less transport of $CO_2$ enriched surface waters into the deep
ocean. Projections indicate a reduced $CO_2$ uptake/sink in the global oceans correlating with a gradual reduction of the
strength of the AMOC (McKinley et al., 2023; Liu et al., 2023). A slowing AMOC impacts ocean biology and solubility
carbon pumps: CMIP5 analysis projects a weakening of the biological pump from surface waters with the largest decreases
in effective $CO_2$ uptake under the strongest warming scenarios (Bopp et al., 2013; Fu et al., 2016; Laufkötter et al., 2016; Liu
et al., 2023). This is in part explained by ocean warming, which result in a decline in PP, also contrbuting to a decrease in



ocean $CO_2$ uptake in some regions (Kwiatokowski et al., 2020), however ocean warming and climate change also leads to sea ice decline, which results in increased PP regionally (Vancoppenolle et al., 2013). The projected climate driven impacts

on the AMOC are yet not seen in the strength of regional components of the AMOC in the northern North Atlantic (Østerhus et al., 2019; Lozier et al., 2019). Also, over the last two decades the estimated AMOC strength at 26°N is dominated by interannual to decadal variability with a marginal significant negative trend (Volkov et al., 2024). North Atlantic ocean and climate variability have the potential to modulate the magnitude of and regional patterns of $CO_2$ sinks.

We aim to understand the sensitivity and drivers of the atmosphere–ocean $CO_2$ flux in an ESM system on

interannual and decadal timescales. Through our analyis we establish which predictors can explain the $CO_2$ flux variability in the five regions of the northern North Atlantic. We define regression models based on physical and dynamic predictors, to explore how well we can predict and understand future $CO_2$ flux variability.

## 1.1 Approach

The aim of this work is to understand how simulated interannual to decadal variability can help building confidence in future

changes and a better understanding of uncertainties related to ocean processes and variability. We analyse ensemble data from 10 historical simulations with the EC-Earth3-CC global climate model with interactive ocean biogeochemical cycling. To assess the drivers of the interannual and decadal $CO_2$ flux variability in the northern North Atlantic we initially establish an overview of the relationship between the $CO_2$ flux variability and parameters that could potentially drive this variability across the northern north Atlantic and in defined sub regions (Fig. 1b).

Based on the gas transfer equation (Eq. 1-3) we expect SST, sea surface salinity (SSS), $\Delta pCO_2$, wind and sea ice concentration (SIC) to be controlling physical parameters driving the $CO_2$ flux variability, however we also define and explore dynamic drivers such as mixed layer depth (MLD) and sea surface height (SSH) as direct indicators of regional circulation changes. For each region, we integrate the $CO_2$ flux and define high and low $CO_2$ flux years at two timescales. We construct ensemble mean composite 2D maos of the $CO_2$ flux by subtracting the low flux years from the high flux years

and by combining all ensemble members. We use the high and low flux years defined for the $CO_2$ flux to also construct composite maps of other key physical parameters.

Informed by the composite maps and identified controls of the $CO_2$ flux variabillity, we define ten indexes that can be correlated with the $CO_2$ flux timeseries. The indexes are based on the ocean circulation strength, AMO, SIC variability, NAO, AMOC strength, wind variability, MLD and SSS variability, the specific definition is described in section 3.

Lastly, in order to understand how much of the $CO_2$ flux variability that can be explained and described by physical variables and indexes alone, we set up simple linear multivariable regression models for each region and discuss differences in explained variability across interannual and decadal timescales.



## 1.2 EC-Earth3-CC

EC-Earth3-CC is the carbon cycle version of the EC-Earth3. EC-Earth3 is an Earth system model, developed by the EC-Earth consortium (https://ec-earth.org), and contributes to the Coupled Model Intercomparison Project (CMIP). EC-Earth3 comprises several model components describing atmosphere, ocean, sea ice, land surface, dynamic vegetation, atmospheric composition, ocean biogeochemistry and the Greenland Ice Sheet. EC-Earth3 consists of the atmosphere model IFS, land surface module HTESSEL and the ocean model NEMO3.6 (Döscher et al., 2022). Coupled to NEMO3.6 is PISCES-v2, a biogeochemical model simulating marine biological productivity and the biogeochemical cycling of carbon and the main nutrients (Döscher et al., 2022; Aumont et al., 2015).

EC-Earth3-CC describes the carbon cycle and is used for the Coupled Climate-Carbon Cycle Model Intercomparison Project (C4MIP; Jones et al., 2016). The configuration allows for simulations with emissions forcings, (however, this function is not used for this study) and the $CO_2$ flux is calculated from and proportional to the difference in partial pressure of ($\Delta pCO_2$) between the atmosphere and the surface of the ocean (Döscher et al., 2022).

EC-Earth3-CC runs with PISCES-v2 (Pelagic Interactions Scheme for Carbon and Ecosystem Studies volume 2), a biogeochemical model that simulates the nutrient cycle and the inorganic and organic carbon cycle. Primary productivity is computed based on the availability of the main nutrients (P, N, Si, Fe), with a constant Redfield ratio (P / N / C, 1 / 16 / 122) (redfield ratio from Takashi et al., 1985; Aumont et al., 2015). Air-sea gas exchange for carbon dioxide is parameterised from Wanninkhof (1992), updated in Wanninkhof (2014) to Eq 1:

$$F = k \, K_0(pCO_{2w} - pCO_{2a}) \, ,$$

where F is the flux (mass area$^{-1}$ time$^{-1}$), k is the gas transfer velocity (lenght time$^{-1}$) and $K_0$ is the solubility (mass volume$^{-1}$ pressure$^{-1}$), which is dependent on water temperature and salinity, $pCO_{2w}$ and $pCO_{2a}$ (pressure) are the partial pressures of $CO_2$ in equilibrium with surface water and the above lying air, respectively. The gas transfer velocity is dependent on wind speed according to Eq. 2:

$$k = 0.251 \, U^2 \, \left(\frac{Sc}{660}\right)^{-0.5}$$

where U is the wind speed, Sc is the Schmidt number. The Schmidt number is dependent on water temperature. Furthermore, the $CO_2$ flux is dependent on the SIC in the region, as no exchange is allowed between the atmosphere and sea water across sea ice Eq. 3:

$$kgCO_2 = k'^{gCO_2} \times (1 - \%_{ice})$$

where $\%_{ice}$ is the concentration of sea ice which varies between 0 and 1 (Döscher et al., 2022, Aumont et al, 2015).

The present day (1982-2014) averaged air-sea $CO_2$ flux shows a general flux into the ocean in the North Atlantic and Nordic Seas (positive $CO_2$ flux, Fig 1a). Observations (Figure 21b from Döscher et al., 2015, based on observational data from Landschutzer et al., 2016) show a similar pattern of a positive flux ($CO_2$ uptake in the ocean) in the North Atlantic and Arctic region, and a negative flux (outgassing to the





atmosphere) off the western coast of Africa across the Atlantic and off the coast of South America. However, observations generally show a smaller air-sea $CO_2$ flux than simulated in EC-Earth3-CC. Döscher et al. (2015) shows that the EC-Earth3-

CC air-sea $CO_2$ flux is in general overestimated in the high northern latitudes (>50°N) in the northern hemisphere, and underestimated in the high latitudes in the southern hemisphere. However, the general distribution of the air-sea $CO_2$ flux is well represented in EC-Earth3-CC.

The sea ice extent of EC-Earth3 (and therefore also EC-Earth3-CC) is in general overestimated in the Nordic Seas (NS) and NA compared to observations (Döscher et al., 2022; Tian et al., 2021, Fig 1b). As no exchange between the

atmosphere and seawater is allowed across sea ice in EC-Earth3, an overestimation of the sea ice extent will inevitably affect the $CO_2$ flux. The global pattern of the $CO_2$ flux and the overestimation of the sea ice extent emphasises the importance of sea ice and high latitude processes in the global climate system (Fig. 1a&b).

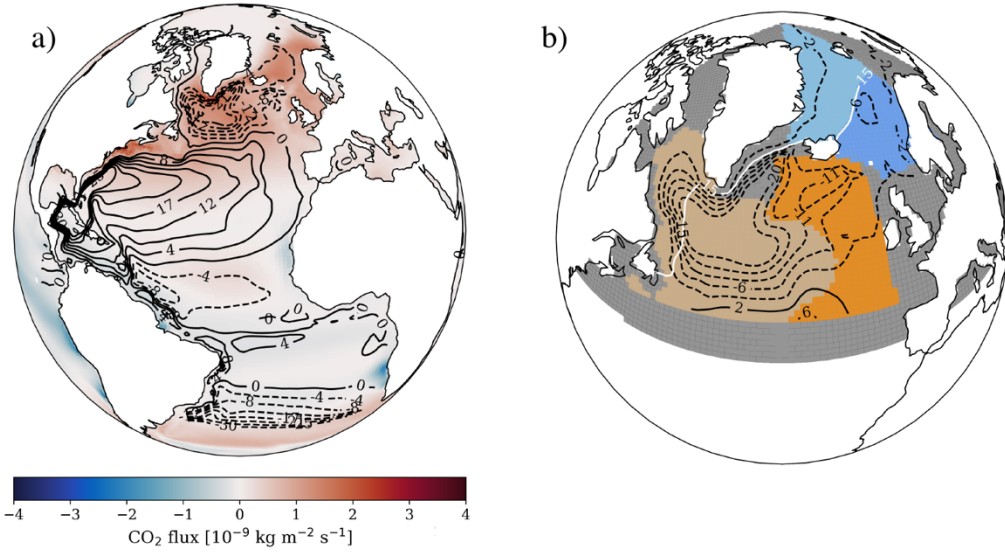

**Figure 1: a) Atmosphere–ocean $CO_2$ flux averaged over the period 1982-2014 for EC-Earth3-CC. Black lines show the mean**
**barotropic streamfunction of the SPG circulation of ten ensemble members in Sv. Full line indicating a positive values, dashed line indicating negative values. Closed dashed lines show the mean extent of the SPG. b) showing the study regions: NA covering all regions (dark grey) SPW (beige), SPE (orange), NSE (blue), NS covering NSE as well (light blue), global average 15% SIC line (white contour) indicating the boundary between sea ice covered regions and sea ice free areas.**

## 2. Data and methods

The analysis is based on EC-Earth3-CC historical runs 1850-2014 from all available ensemble members on ESGF (*https://aims2.llnl.gov/search/cmip6/*) (ten ensemble members: r1i1p1f1, r4i1p1f1, r6i1p1f1, r7i1p1f1, r8i1p1f1, r9i1p1f1, r10i1p1f1, r11i1p1f1, r12i1p1f1, r13i1p1f1). We make use of monthly mean fields of the parameters shown in Table 1. The data is preprocessed pointwise for analysis on two seperate timescales: decadal and interannual. For the decadal timescale a 10 year low-pass filter (using CDO, filtering the data in the frequency domain: Schulzweida et al., 2012) has been applied to



the data. Following the low-pass filtering the data has been linearly detrended to remove the anthropogenic forced changes due to increases in atmospheric greenhouse gases and to be able to focus on the internal climate and ocean variability of the North Atlantic. The atmosphere–ocean $CO_2$ flux (and $\Delta pCO_2$) shows a modest linear increase from 1850–1950 and then a transition to a steeper increase until 2015 (see also section 2.2, Fig. 2). Therefore, in this case a two step piecewise linear fit has been used to detrend the dataset. The model dataset has been detrended along the time axis for every grid cell, allowing

for analysing the detrended data on a regional scale. Other parameters have been detrended using a linear fit. We choose not to apply any detrending to the streamfunction data at grid point level. A second preprocessed dataset of interannual variability is prepared by subtracting the low-pass filtered timeseries from the original model data. Also here, the interannual $CO_2$ flux and $\Delta pCO_2$ dataset has likewise been detrended using a two-step linear fit, and other parameters using a linear fit.

**Table 1**

Parameters from EC-Earth3-CC

| Parameter | Output variable name | Form of detrending | Unit |
|---|---|---|---|
| $CO_2$ flux | fgco2 | two-step linear | kg m$^{-2}$ s$^{-1}$ |
| $\Delta pCO_2$ | dpco2 | two-step linear | Pa |
| Sea surface temperature (SST) | tos | linear | °C |
| Sea surface salinity (SSS) | sos | linear | |
| Wind | sfcWind | linear | m s$^{-1}$ |
| Mixed layer depth (MLD) | mlotstmax | linear | m |
| Sea surface height (SSH) | zos | linear | m |
| Sea ice concentration (SIC) | siconc | linear | % |
| Pressure at sea level | psl | linear | Pa |
| AMOC | msftyz | None | kg s$^{-1}$ |
| Barotropic mass streamfunction | msftbarot | None | kg s$^{-1}$ |

**2.1 Regions**

This study looks into the $CO_2$ flux variability in multiple regions in the North Atlantic (Fig. 1b), to investigate the regional differences in the relationship between the $CO_2$ flux and the physical and dynamical processes. The geographic regions are defined considering the model representation of oceanographic domains. The North Atlantic (NA) region covers all of the ocean in the region 40°N - 90° N, -78°W - 45° W, with avereage ocean $CO_2$ uptake. The Nordic Seas (NS) is defined

roughly where warmer subpolar North Atlantic waters in the model meets waters of the Nordic Seas at the latitude of the Greenland Scotland Ridge and defined as the region 60°N - 90° N, 28°W - 18°W. The Nordic Seas east (NSE) is defined as





the NS, but only where model climatology has less than 15% sea ice and as such, representing the sea ice free part of the NS, allowing for unbiased investigation of the dynamics not related to the overestimated sea ice extent. The Subpolar region of the North Atlantic is divided into two oceanic domains: the Subpolar West (SPW) and the Subpolar East (SPE). The SPW is defined as where the annual mean SSS is <34.2 within the region 45°N - 60°N, 52°W -10°W, the part of the subpolar gyre recirculating watermasses modified by low salinity polar outflow and melting sea ice. The SPE is defined as SSS >34.2 within the region 45°N - 64°N, 52°W - 10°W in the path of the North Atlantic current. The SPE, like the NSE, represents a region with no sea ice cover (climatological SIC <15%).

## 2.2 $CO_2$ flux variability

**Table 2**

Variability (standard deviation, std) of ensemble mean $CO_2$ fluxes for F (model output), $F_{calc}$ (calculated flux) in interannual (int) and decadal (dec) timescales

|  | F [$10^{-11}$ kg m$^{-2}$ s$^{-1}$] | | $F_{calc}$ [$10^{-11}$ kg m$^{-2}$ s$^{-1}$] | | Area of region [$10^{12}$ m$^2$] |
|---|---|---|---|---|---|
| Std | Int | Dec | Int | Dec | |
| NA | 3.22 | 2.16 | 2.92 | 2.39 | 11.27 |
| NS | 5.35 | 5.87 | 5.05 | 5.56 | 0.96 |
| NSE | 8.23 | 6.99 | 7.08 | 5.93 | 0.27 |
| SPW | 8.22 | 5.49 | 7.52 | 5.28 | 2.86 |
| SPE | 6.85 | 5.84 | 5.96 | 4.97 | 1.04 |

Archived climate model data typically available from CMIP6 on the ESGF nodes typically consists of monthly mean fields of physical, biological and chemical parameters. This includes surface air sea $CO_2$ fluxes from EC-Earth3-carbon cycle as well as ocean properties. The level of reproducibility of simulated integrated fluxes from other parameters (Eq. 1-3) (F) and in particular their variability across time-scales from archived averaged monthly data will constitute an upper limit for our ability to explain from physical quantities the model variability. Figure 2 is also shows the time evolution of $F_{calc}$ ($CO_2$ flux calculated according to Eq 1-3) of a selected ensemble member (r4i1p1f1) and for the NS, NSE and SPE region only. To accurately reproduce the integrated flux, $F_{calc}$ is computed using a scaled wind speed. Since $F_{calc}$ is derived from monthly mean values of all parameters, some of the short-term variability in dynamic factors—particularly wind speed—is smoothed out, leading to a lower mean wind speed compared to the daily mean used in EC-Earth3-CC. Hughes et al. (2012) discusses a averaging-realated bias linked to ocean surface flux calculations and show that the use of monthly mean fields can introduce a bias into the mean flux estimates (also discussed by Esbensen and Reynolds, 1981; Simmonds and Dix 1989; Gulev, 1994; 1997; Josey et al., 1995; Zhang 1995; Esbensen and McPhaden 1996; Simmonds and Keay 2002). Given that Eq. 2 shows a nonlinear dependence of $CO_2$ flux on wind speed ($U^2$), even small-scale wind variability can have a significant impact on flux calculations. To



account for the reduced wind variability in the monthly mean fields and ensure consistency with the original flux calculation,
a scaling factor of 1.2 is applied to the wind speed when computing $F_{calc}$. The scaled wind is only used for this calculation and
not in any of the other analysis. It is also expected that the $CO_2$ flux variability is dependent on SST and SSS variability,
however the effect is too small to be considered important in these calculations, and the SST and SSS components of Eq 1-3
is therefore not scaled. The absolute level of the $F_{calc}$ compared to the model flux is well captured including the trend. For this
ensemble, the correlation between the estimated flux and the model output in SPE exceeds 0.9 ($R^2$) with a regression factor
close to unity (1.1). We find comparabale/significant skill in reproducing fluxes for the shorter timescale (not shown).

      Figure 2 shows the average (weighted mean) $CO_2$ flux timeseries F for each region with interannual variability
removed. The antroprogenic trend is evident in all regions including the accelerated trend after 1950. Difference between
regions are evident with NSE showing the largest $CO_2$ uptake per unit area. Strong decade-scale variations do modulate
regionally the accelerated trend after 1950 emphasizing the importance of ocean variability. Removing the anthropogenic trend
and isolating interanual and decadal variability reveals i) relatively comparable levels of variability across regions and ii) a
stronger variability on short timescales (Table 2). Ensemble mean results indicates that NSE, which is also the smallest area,
is subject to strongest variablity on both timescales, followed by the SPW (Table 2).

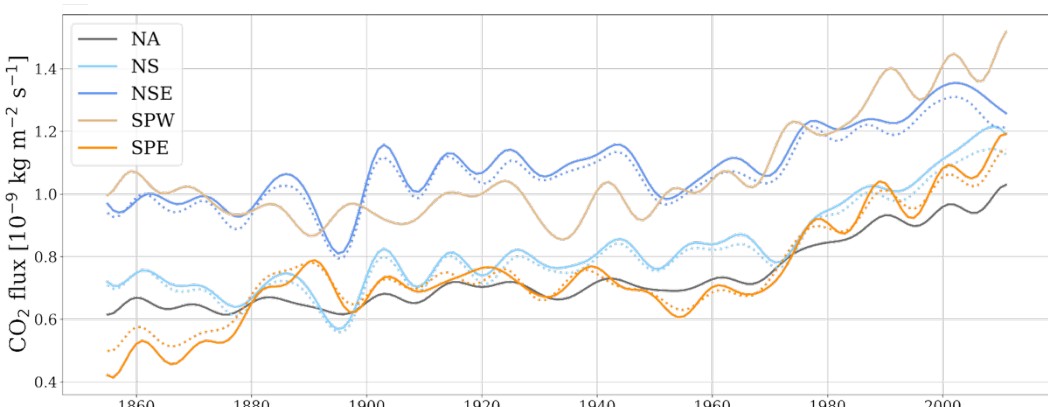

**Figure 2: Decadal variability in the simulated $CO_2$ flux (F, solied line) and calculated flux ($F_{calc}$, dotted line) for ensemble member**
**r4i1p1f1. Here the weighted mean is shown to compare the fluxes and variability across regions. $F_{calc}$ only showed for NS, NSE, SPE**







## 3. Results

### 3.1 The variability between timescales and regions

To visualise the patterns of $CO_2$ flux variability across timescales and regions, we create composite maps of the $CO_2$ flux and the parameters we expect to be important drivers of the variability. These parameters include mixed layer depth (MLD), sea
surface height (SSH), SIC, SST, SSS, $\Delta pCO_2$ and wind. After detrending the data as described above, we define high and low $CO_2$ flux years in the different study regions based on the mean standard deviation for all the ten members (+/-) one std from the regional mean. The years are defined and selected for each member for all the parameters, where after the difference of the ensemble mean high minus low flux years is shown in composite maps (Fig. 3-7).

          The North Atlantic region covers all the other regions. The $CO_2$ composite maps show a widespread homogeneous
pattern of $CO_2$ uptake, intensified across the subpolar region (Fig. 3) and with regional differences. This includes the Irminger and Iceland Basins, where fluxes are moderated and locally reversed. On decadal timescales, positive values (increased $CO_2$ uptake) are confined to areas near the sea ice edge indicating that this exert an important control on North Atlantic average $CO_2$ flux variability at longer timescales (Fig. 3b). On interannual timescale, the widespread $CO_2$ uptake correlates with a widespread pattern of stronger winds and colder SST, indicating that these parameters are potential drivers
of the $CO_2$ flux variability on shorter timescales. This is not the case for decadal variability where only positive wind speed project on regions of enhanced $CO_2$ flux, temperature generally warm and act to reduce solubility. A positive $\Delta pCO_2$ pattern correlates with the positive $CO_2$ flux anomaly and negative SST anomaly (Fig. 3a). On the decadal timescale, a negative SSH anomaly, indicating a strengthening of the subpolar gyre, correlates with the positive $CO_2$ flux anomaly in the subpolar gyre region, which suggest that the strength of the subpolar gyre also could be a potential dynamical driver of the $CO_2$ flux
variability, modifying key parameters like SST and SIC. The $CO_2$ flux variability pattern in the North Atlantic appears relatively homogenous, while its correlations with other parameters are less distinct, except for SST on short timescales and SIC on long timescales. This suggests that multiple dynamics may be influencing $CO_2$ flux variability. To better understand these influences, we will analyse smaller, well-defined regions to identify the key drivers of $CO_2$ flux variability.






## North Atlantic



**Figure 3 Anomalies of the high-low years of the dynamic parameters compared to the CO₂ flux anomaly in the NA. a) showing interannual timescale and b) showing decadal timescale. Full and dashed lines on top of the SSH anomaly shows the barotropic mass streamfunction anomaly and dashed line on top of SIC anomaly shows the 15% SIC climatology (mean of ten ensemble members on both timescales).**

Focusing in on the the Nordic Seas region, we see a widespread and homogeneus pattern of CO₂ uptake on interannual

timescale (Fig. 4), on short timescales confined to the Nordic Seas emphasising the role of regional dynamical systems and

need for a regional analysis. The Nordic Seas shows the strongest flux anomalies across all the regions, in particular on long

timescales. On decadal timescales, the CO₂ uptake is confined to an area of varying SIC in the Nordic Seas with dynamical





consistent patterns also in the SPG region, indicating that SIC is the main driver of the $CO_2$ flux variability in this region on
longer timescale (Fig. 4a). Positive SST anomalies reaching 5°C on decadal scale would conteract the effects of sa ice
variability. On the shorter timescale, the $CO_2$ uptake is enhanced  across most of the Nordic Seas region, however, on
decadal scale a small part of the region shows $CO_2$ outgassing (Fig. 4b), likely linked to the strong warming of the composite
and reduced $\Delta pCO_2$. Again, the positive $CO_2$ flux anomaly is perfectly confined to areas within an area of decrease in SIC,
indicating that SIC is the main driver of $CO_2$ flux variability on the shorter timescale as well. The decrease in SIC is greater
in the Nordic Seas composite maps compared to the North Atlantic maps. On both timescales stronger winds correlates with
the increased $CO_2$ uptake, similar to the North Atlantic region. However, the wind anomaly is stronger on the longer
timescales compared to the shorter timescales. Following a decrease in SIC we see a deepening of the MLD due to increased
convection, increasing the mixing and allowing for an increase in air-sea $CO_2$ flux. However, the small scale and spatial
pattern associated with enhanced mixing is not obviously reflected in the patterns of $CO_2$ uptake. Also, from the spatial
correlation of patterns, it is clear that $\Delta pCO_2$ changes alone cannot explain the $CO_2$ flux variability.





## Nordic Seas



**Figure 4 Anomalies of the high-low years of the dynamic parameters compared to the CO₂ flux anomaly in the NS. a) showing interannual timescale and b) showing decadal timescale. Full and dashed lines on top of the SSH anomaly shows the barotropic mass streamfunction anomaly and dashed line on top of SIC anomaly shows the 15% SIC climatology (mean of ten ensemble members on both timescales).**

The Nordic Seas East region represents the sea ice free part of the Nordic Seas (less than 15% SIC) and is in the path of the North Atlantic current crossing the Greenland Scotland Ridge. The composite maps of high minus low $CO_2$ flux years show an increased $CO_2$ uptake (Fig. 5) confined to the NSE region highlighting that oir regionalisation is relevant and allow us to




isolate a dynamical regime. The region is confined by the 15% SIC line, and show a widespread positive $CO_2$ flux anomaly. Figure 5a illustrates that the interannual variability shows an increase in wind correlating with a deepening of the MLD and an increase in the air-sea $CO_2$ flux. These patterns are more pronounced for the NSE than for the NS region. A decrease in SSH indicates a strengthening of the Nordic Seas gyre circulation, the eastern part, correlating with the increase in MLD and

$CO_2$ flux indicating a dynamic driver of the $CO_2$ flux variability. Figure 5b shows a confined pattern of $CO_2$ uptake, correlating with a confined pattern of stronger wind, increased MLD as a result of a strengthening of the Nordic Seas Gyre (NSG; shallowing of the SSH) and increased SSS. Compared to the North Atlantic region, the Nordic Seas East give a clearer indication of the driving forces behind the $CO_2$ flux variability.



## Nordic Seas East

**Figure 5 Anomalies of the high-low years of the dynamic parameters compared to the CO₂ flux anomaly in the NSE. a) showing interannual timescale and b) showing decadal timescale. Full and dashed lines on top of the SSH anomaly shows the barotropic mass streamfunction anomaly and dashed line on top of SIC anomaly shows the 15% SIC climatology (mean of ten ensemble members on both timescales).**

In the western part of the subpolar region, the CO₂ flux spatial pattern is strikingly similar to the pattern we can construct for the entire North Atlantic on both timescales (Fig. 3 and 6). This can in part be expected from the vast area of the region hereby dominating the area-averaged fluxes and composites. Key differences with the NA region includes the Irminger Sea where also SSH patterns indicate a dynamical separation on interannual scales. On the interannual timescale the region





shows $CO_2$ uptake within the full region (Fig. 6a), whereas the decadal $CO_2$ flux shows both positive and negative anomalies within the region (Fig. 6b). In general all the parameters show similar patterns to the North Atlantic on both timescales.

However, on the interannual timescale we see a more clear and confined signal from SST and SSS compared to the North Atlantic. A negative SST anomaly correlating with increased $CO_2$ uptake in the full region, whereas a positive SSS anomaly in most of the region also correlates with the $CO_2$ flux variability, emphasising how the SSS can work as an indirect driver or indicator. On the decadal timescale we see a clear and confined signal in the indicated drivers of the $CO_2$ flux compared to the North Atlantic region. Here, we see that the $CO_2$ uptake is confined within the region of decreasing SIC, indicating that

SIC forces a main control on the $CO_2$ flux variability in the region. Furthermore, we see a greater strengthening of the subpolar gyre as indicated by the decrease in SSH on both timescales, however the strongest signal is seen on the longer timescale, where we also see a shift towards the east as expected in a strong SPG phase. The strengthening of the subpolar gyre results in a deepening of the MLD and introduces more saline water into the system, explaining the correlation between a positive SSS anomaly and increased $CO_2$ uptake. As in the North Atlantic region, we see opposite anomalies for the SST

on the two timescales. A negative SST anomaly correlates well with a positive $CO_2$ flux anomaly on the short timescale, whereas on the longer timescale, we see a positive SST anomaly correlating with a positive $CO_2$ flux anomaly.





**Figure 6 Anomalies of the high-low years of the dynamic parameters compared to the CO₂ flux anomaly in the SPW. a) showing interannual timescale and b) showing decadal timescale. Full and dashed lines on top of the SSH anomaly shows the barotropic mass streamfunction anomaly and dashed line on top of SIC anomaly shows the 15% SIC climatology (mean of ten ensemble members on both timescales).**





Lastly, we look at the eastern part of the subpolar region (Fig. 7). The composite maps show patterns of increased $CO_2$ uptake, which is regionally well confined to the eastern subpolar gyre and not apparently analogue to the composite patterns derived for SPW. This includes also the North Atlantic region. On the interannual timescale SSH reflects what may best be characterized as a relative blocking of the path of the North Atlantic current (a closed gyre system building up sea surface

height). On the decadal timescale this blocking cell is located further south, outside of the SPE region, but in both cases result in cooling of the wider subpolar region as well as freshening. Despite these basin wide linkages, the $CO_2$ flux changes are regionally exaggerated in the SPE. Not surprisingly, the dynamical changes – a blocking of the path of the North Atlantic current – are also characteristic of the composite maps for the NSE region (Fig. 5). Despite this apparent similarity, the $CO_2$ flux patterns are in both cases not extending across the Greenland Scotland Ridge. This may be explained by phase

differences in the impacts.

       The spatial pattern of positive $CO_2$ flux anomaly correlates with a shallowing of the MLD on the interannual timescale. This cannot directly be linked to increased uptake, on the contrary as described in section 1 and 3.1, as a shallowing of the MLD is related a less ventilated water coloumn, and thereby a decrease in the $CO_2$ uptake. This corelation will be discussed further in the next section. Also on the interannual timescale, a positive wind anomaly, and a negative SST

and SSS anomaly also correlates with the positive $CO_2$ flux anomaly. On the decadal timescale, the signal from the MLD and wind is less clear, however, a negative SST and SSS anomaly correlates with the increases $CO_2$ uptake in the region.







**Figure 7 Anomalies of the high-low years of the dynamic parameters compared to the CO₂ flux anomaly in the SPE. a) showing interannual timescale and b) showing decadal timescale. Full and dashed lines on top of the SSH anomaly shows the barotropic mass streamfunction anomaly and dashed line on top of SIC anomaly shows the 15% SIC climatology (mean of ten ensemble members on both timescales).**





### 3.2 Ocean indexes

Following the qualitative discussion of the relationship between the $CO_2$ flux variability and the dynamical parameters on different timescales, we define integrated quantities and make use of ocean indicators such as the subpolar gyre index to understand statistical significance of the identified relationships between the $CO_2$ flux and ocean variability (Table 3).

The Subpolar Gyre index (SPGi) is here defined simply as the minimum SSH of the NA region. This differs from other definitions (Hátún et al. 2005, Berx and Payne et al., 2017) but captures the main variability of the Earth System Model

analysed. The index is inverted, meaning that a positive (negative) anomaly indicates a strong anticyclonic (weak) SPG. The subpolar gyre index describes not only the state of the subpolar region but from observations also shown to characterize variability in water mass properties at the gateways to the Nordic Seas (Hátún et al 2017). As such, the SPGi may be linked to $CO_2$ flux variability directly by describing the extent of the characteristic colder and fresher subpolar upper ocean water masses and indirectly through correlations with other drivers such as variability in sea-ice margins, wind stress and ocean

mixing locally and remote (Nordic Seas). From correlation coefficients (zero lag) between the SPGi with the $CO_2$ flux anomaly for the different regions (NA, NS, NSE, SPW, SPE), presented in Table 3, it is clear that the SPGi does not explain the $CO_2$ flux variability on the interannual timescale very well, though reassuringly the highest correlation are found for the SPW region. The SPGi correlates better on the decadal timescale, with an ensemble mean of 0.31 and 0.44 in the NA and SPW respectively. Several ensemble members show a correlation coefficient of 0.4 or higher in both regions, but a few

ensemble members show very low correlation, resulting in reduced ensemble mean values. The SPE shows a large spread in individual ensemble correlations with both positive and negative correlations, resulting in an ensemble mean of -0.24 and -0.17 (interanual and decadal, respectively). The low degree of correlation on both timescales is expected in part due to the finding from the composite map analysis of a more regional SSH (and streamfunction) 'blocking pattern' of the North Atlantic Current.

The Nordic Seas Gyre index (NSGi) is defined as the minimum SSH in the NS region, similar to the SPGi. The index is inverted, meaning that a positive value indicates a strong anticyclonic gyre circulation, whilst a negative value indicates a weakening of the gyre circulation. The NSGi will be modulated by changes in wind stress curl and also not independent of other possible indices defined on the basis of MLD and SIC. We have no a-priori expectation of a strong relation between the NSGi and $CO_2$ flux in other regions. The NSGi correlates well with the $CO_2$ flux variability in the NS

on the decadal timescale (0.54, Table 3), but also on the interannual timescale (0.30, Table 3). A strong (weakened) NSGi results in an increased (decrease in) $CO_2$ uptake in the ocean. Hátun et al. (2021) shows that a strong NSG circulation results in an uplifted thermocline, which results in more ventilation of deeper water. The ventilation of deeper water could lead to an increased air-sea $CO_2$ flux, and work as a driver for the increased $CO_2$ uptake in the NS. The composite maps shows a strenghtening of the NSG (deepening of the SSH) on both timescales, however a greater strengthening on decadal timescale,

indicating, that a stregthening of the NSG could result in an increased atmosphere–ocean $CO_2$ flux.



The AMO index is defined as the anomaly of the mean SST in the NA region. For the regions within the NA, the index is defined as a SST index, which is the anomaly of the mean SST within the region. We have chosen to use the SSTi for the smaller regions, to investigate the direct realtionship of SST and $CO_2$ flux variability wihtin the regions. A positive AMO/SSTi phase describes a warming of the North Atlantic. As higher SST's increase the $pCO_2$ in the ocean due to lower

solubility, the AMO should be negatively correlated to the $CO_2$ flux. However, Breeden and McKinley (2016) shows how a positive AMO phase results in a decrease in DIC in the SPG, which decreases the solubility and increases the $fCO_2$ (and the $pCO_2$). On the decadal timescale we see that the $CO_2$ flux variability in NS and SPW region correlates positively (0.36 and 0.29, Table 3) with the SSTi, which can be related to the dynamics explained by Breeden and McKinley (2016). Also, an increase in temperature in the NS would result in a decrease in SIC, allowing for an increase in $CO_2$ flux. Breeden and

McKinley (2016) states that SST and the solubility of $CO_2$ in the ocean still affects the $pCO_2$ in the ocean (and thereby the air-sea $CO_2$ flux), but that the biogeochemical dynamics, such as DIC are of a stronger amplitude. The NSE and SPE regions shows an inverted correlation between the $CO_2$ flux variability and the SSTi on both timescales (NSE: -0.12 and -0.41 SPE: -0.15 and -0.38, interannual and decadal), indicating that the temperature effect on the $CO_2$ solubility in the surface ocean dominates the $CO_2$ flux variability in these regions (Table 3). In the NA the degree of correlation varies a lot between the

ensemble members from a negative/weak correlation to a stronger correlation, which makes it difficult to say anything in general about the full NA region.

The sea ice index (SIi) is defined as the SIC anomaly in the specified regions. An increase in SIC results in a decrease in $CO_2$ flux into the ocean in case no other factors are changing. In the NS the SICi in general has a strong negative correlation with the decadal $CO_2$ flux anomaly, all of the members showing a correlation coefficient above 0.50, resulting in

an ensemble mean of -0.64 (Table 3). The interannual variability in the NS shows a lower correlation, with an ensemble mean of -0.36 (Table 3). On both timescales, the SPW region shows a weaker negative correlation compared to the NS (-0.04 and -0.36, interannually and decadal), as expected due to a smaller degree of sea ice extent in the SPW. However, it is clear that the SIC is still dominating the $CO_2$ flux variability in this region. The NA region shows the weakest correlation between the SICi and the $CO_2$ flux variability, which migh be explained by the fact that the sea ice covered region of the NA

is small (-0.02 and -0.26 interannually and decadal). The SICi has not been applied to the NSE and SPE regions, as they are more or less sea ice-free.

The NAO index is defined as the atmospheric surface pressure level anomaly between Iceland (65° N, 20° E) and the Azores (40° N, 25° E) in the winter months (djfm; Hurrel 1995). The correlation between the NAO and $CO_2$ flux is in general strongest on the interannual timescale, showing a correlation coefficient of around 0.3 for most regions, except SPE

that shows 0.12 and 0.35 interannual and decadal, respectively. On the decadal timescale the ensemble mean correlations are weaker, however in particular in the ice-free SPE and NSE and in the NA some ensemble members show a correlation of 0.4-0.5 (Table 3). The NAO and associated wind stress and wind stress curl impacts the ocean dynamics of the region including sea ice distribution which may explain the positive correlations, whereas the direct effects of changes in wind stress on gas transfer would be expected to result in negative correlations for some regions.



435       The AMOC index (AMOCi) is defined as the anomaly of the maximum of the model Atlantic Meridional Overturning Circulation (streamfunction) at 45°N in the top 1000m of the water column.We choose 45°N to align with the boundary of the Subpolar and subtropical gyres and reduce potential time lag, that would follow from using an index defined at lower latitudes. On the interannual timescale, the AMOCi shows a strong negative correlation in the NA and SPW regions (-0.40 and -0.47, Table 3), indicating that a weakened AMOC resulting in less heat transport to the Northern North Atlantic

will result in increased ocean $CO_2$ uptake. In the other regions, the ensemble members shows a low correlation on the short timescale. On the decadal timescale, somewhat surprsinsgly, the AMOCi is not as closely and directly related to surface $CO_2$ exchange with both negative and positive correlation coefficients in all the regions, resulting in a weak average correlation (Table 3).

       The wind index (Wi) is defined as the anomaly of the average near surface (10 m) wind speed in the different

regions. As expected based on Eq. 2, the Wi correlates positively with the $CO_2$ flux variability for all regions independent on timescale. On both interannual and decadal timescale, the Wi shows a high correlation with the $CO_2$ flux in the NA (0.68 and 0.45), NS (0.73 and 0.70) and NSE (0.72 and 0.57, Table 3). However, SPW only show a high correlation on interannual timescale (0.80 and 0.26 decadal), whereas and SPE shows a weaker correlation on both timescales (0.24, 0.26). In the composite maps all the regions show a strong positive wind anomaly, whereas the SPE region shows a weak positive

anomaly on the interannual timescale and a mix of negative and positive anomalies on the decadal timescales, which might explain the weak correlation seen in the region.

       To understand the dynamic process of upper ocean mixing and its affect on the atmosphere–ocean $CO_2$ flux variability a MLD index (MLDi) is defined based on the anomaly of the MLD max value of the different regions. The MLDi shows a strong negative correlation in the SPE (0.53 interannual; 0.29 decadal, Table 3) on both timescales, and a strong

positive anomaly in the NS (0.38 interannual, 0.57 decadal) indicating a complex, indirect and regionaly different linkage. Spatial patterns of MLD show some resemblance to $\Delta pCO_2$ in the SPE with an inverse relation (Fig. 7), which can be explained by shoaling of the mixed layer leading to less ventilation of DIC rich deeper layersand thus increased $\Delta pCO_2$.

       Salinity directly, but weakly influence the gas transfer (Eq. 1), but we choose to define also a SSS index (SSSi) as preliminary analysis showed a clear correlation between PP and SSS in the EC-Earth3-CC output data which can be

exploited to be able to use the analysis from this study on different ESM's that might not include a biogeochemical component in possible future studies. Furthermore, the SSS dynamics also translates to the strength of the SPG, the melting of sea ice and transports of different water masses in the North Atlantic. We have defined the SSSi as the SSS anomaly in the different regions. The SSSi shows the strongest interannual correlation in the SPW (0.33), and the strongest decadal correlation in the NS (0.52) and SPW (0.39). In the composite map of the NS region, we see a strong positive SSS anomaly

correlating with a positive $CO_2$ flux anomaly (Fig. 4). The SSSi shows a strong negative correlation with the $CO_2$ flux in the SPE, which is also evident in the composite maps (Fig. 7).





**Table 3**

Index analysis. Correlation coefficients are the mean of 10 ensemble members in interannual (int) and decadal (dec) timescales. In cursive min and max values for individual members are shown to indicate spread (uncertainty) of the index analysis. In bold parameters used for regression models in section 3.3.

|  |  | NA | | NS | | NSE | | SPW | | SPE | |
|---|---|---|---|---|---|---|---|---|---|---|---|
|  | Corr. Coeff. | Int | Dec | Int | Dec | Int | Dec | Int | Dec | Int | Dec |
| SPGi | mean | 0.04 | **0.31** |  |  |  |  | 0.21 | 0.44 | -0.24 | -0.17 |
|  | *min* | *-0.16* | *0.08* | - | - | - | - | *-0.01* | *0.18* | *-0.42* | *-0.71* |
|  | *max* | *0.16* | *0.69* |  |  |  |  | *0.25* | *0.80* | *0.05* | *0.27* |
| NSGi | mean |  |  | 0.30 | 0.54 | 0.25 | **0.49** |  |  |  |  |
|  | *min* | - | - | *0.13* | *0.41* | *0.11* | *0.31* | - | - | - | - |
|  | *max* |  |  | *0.53* | *0.76* | *0.40* | *0.66* |  |  |  |  |
| AMO /SSTi | mean | -0.27 | 0.19 | 0.09 | 0.36 | **-0.12** | -0.41 | **-0.25** | **0.29** | **-0.15** | **-0.38** |
|  | *min* | *-0.46* | *-0.13* | *-0.05* | *0.03* | *-0.39* | *-0.64* | *-0.42* | *-0.33* | *-0.31* | *-0.83* |
|  | *max* | *-0.04* | *0.56* | *0.28* | *0.59* | *0.19* | *-0.24* | *-0.12* | *0.64* | *0.03* | *0.38* |
| SICi | mean | -0.02 | -0.26 | **-0.36** | **-0.64** |  |  | -0.04 | -0.36 |  |  |
|  | *min* | *-0.09* | *-0.49* | *-0.51* | *-0.84* | - | - | *-0.09* | *-0.68* | - | - |
|  | *max* | *0.08* | *0.07* | *-0.17* | *-0.54* |  |  | *0.13* | *-0.29* |  |  |
| NAO | mean | 0.30 | 0.24 | 0.29 | 0.18 | 0.35 | 0.31 | 0.30 | 0.13 | 0.12 | 0.35 |
|  | *min* | *0.17* | *0.02* | *0.20* | *-0.09* | *0.24* | *0.14* | *0.19* | *-0.19* | *-0.09* | *0.11* |
|  | *max* | *0.46* | *0.47* | *0.37* | *0.30* | *0.44* | *0.45* | *0.46* | *0.30* | *0.32* | *0.57* |
| AMOCi | mean | **-0.40** | 0.20 | 0.11 | 0.26 | 0.17 | **-0.06** | **-0.47** | 0.16 | -0.20 | 0.02 |
|  | *min* | *-0.53* | *-0.08* | *-0.06* | *0.02* | *0.05* | *-0.33* | *-0.54* | *-0.21* | *-0.33* | *-0.42* |
|  | *max* | *-0.32* | *0.49* | *0.28* | *0.47* | *0.34* | *0.14* | *-0.37* | *0.42* | *-0.12* | *0.45* |
| Wi | mean | **0.68** | **0.45** | **0.73** | **0.70** | **0.72** | **0.57** | **0.80** | **0.26** | **0.24** | 0.26 |
|  | *min* | *0.57* | *0.26* | *0.66* | *0.57* | *0.59* | *0.40* | *0.71* | *0.17* | *0.04* | *0.01* |
|  | *max* | *0.78* | *0.56* | *0.80* | *0.83* | *0.78* | *0.66* | *0.89* | *0.80* | *0.41* | *0.43* |
| MLDi | mean | 0.02 | 0.28 | **0.38** | **0.57** | **0.33** | 0.01 | 0.14 | **0.38** | **-0.53** | **-0.29** |
|  | *min* | *-0.14* | *-0.03* | *0.18* | *0.36* | *0.15* | *-0.36* | *-0.07* | *-0.12* | *-0.67* | *-0.46* |
|  | *max* | *0.13* | *0.59* | *0.65* | *0.80* | *0.61* | *0.50* | *0.31* | *0.61* | *0.03* | *-0.14* |
| SSSi | mean | 0.21 | **0.31** | 0.22 | 0.52 | 0.08 | -0.24 | 0.33 | 0.39 | -0.29 | **-0.37** |
|  | *min* | *-0.19* | *-0.05* | *0.02* | *0.33* | *-0.17* | *-0.49* | *0.04* | *0.03* | *-0.37* | *-0.71* |
|  | *max* | *0.33* | *0.55* | *0.40* | *0.63* | *0.31* | *-0.12* | *0.45* | *0.54* | *0.03* | *0.04* |





### 3.3 Regression analysis

A regression model has been set up for each of the regions and both timescales, based on the findings from the composite maps and the index correlation analysis. The predictors chosen for the regression models have been chosen to provide the best possible regression model, not only focusing on the linear relationship between the predictors and the $CO_2$ flux as shown in the index analysis (Table 3). Some predictors might show a weak correlation coefficient, but contribute to a higher degree of explanation in combination with other predictors. For instance the NAO index is not considered for the regression models, as it to some degree represents the same dynamics as the wind index, and therefore does not contribute to a higher degree of explanation for the regression models. In the following the descriptions of the regression models is based on the mean $R^2$ value of the 10 ensemble members for each region (Table 4). The regression models for the interannual variability shows slightly higher correlation/explanation for the $CO_2$ flux variability in the different regions compared to the decadal variability. In general, the NA region shows the lowest degree of correlation on both timescales, emphasising the need to divide the region into smaller regions.

The North Atlantic region interannual $CO_2$ flux can be explained by 0.48 $R^2$ by the variability in AMOC and wind speed (Table 4). The spatial patterns show a strong, homogeneous positive wind anomaly correlates well with a positive $CO_2$ flux anomaly in the NA (Fig. 3). Furthermore, the index analysis show that both the AMOCi and Wi correlates well with the $CO_2$ flux variability (0.40 and 0.68, respectively, Table 3). The wind speed is expected to be driving the $CO_2$ flux variability, but the AMOCi represents a more dynamic control on the variability. The decadal timescale regression model includes predictors as SPGi, SSSi and Wi. Here, the $R^2$ value is 0.37 (Table 4). The composite map does not show a strong spatial pattern between the predictors and the $CO_2$ flux variability, which might explain the lower $R^2$ value. The predictors chosen for the decadal regression model aligns with the indexes with the greatest correlation as seen in Table 3.

The Nordic Seas region shows a good degree of explanation (0.62 and 0.58 for the interannual and decadal timescales respectively, Table 4) based on MLDi, SICi and Wi. The sea ice concentration works as a main driver for the $CO_2$ flux variability in the NS as discussed earlier and seen in Fig. 4. Furthermore, a strong positive signal in the MLD anomaly correlates with the $CO_2$ flux anomaly, indicating that dynamical processes, such as strengthening of the NSG (a deepening of the SSH) and increased convection related to disappearing sea ice also controls the $CO_2$ flux variability in the NS. The index correlation analysis also indicates the wind stress as a controlling parameter in this region in both timescales (Table 4). The fact that the regression model is based on the same parameters for both timescales indicates that the processes driving the $CO_2$ flux variability is stable across timescales for this region.

The Nordic Seas East region shows a high degree of explanation as well (0.68 and 0.53 interannual and decadal respectively, Table 4). The interannual predictors being MLDi, SSTi and Wi and the decadal NSGi, AMOCi and Wi. The interannual and decadal variability is well explained by the spatial patterns in the region (Fig. 5). However, the index analysis show a weak linear correlation between the predictors SSTi and AMOCi and the $CO_2$ flux on the interannual and decadal timescale respectively (Table 3). The low correlation coefficient in the index analysis does not however rule out, that



the SST and AMOC contribute to the variability combined with other predictors, as shown in the regression models for the region. The NSE regression models show that dynamical processes such as ocean mixing and circulation drive the $CO_2$ flux

variability within the region in addition to physical parameters such as SST and wind, which is expected from the gas transfer equations (Eq. 1-3).

The western part of the Subpolar region shows the next best degree of correlation on the interannual timescale (0.68 interannual and 0.50 decadal, Table 4). The regression model setup based on the interannual variability entails AMOCi, SSTi, and Wi and show a $R^2$ value of 0.68. Based on MLDi, SSTi and Wi the regression model for the variability on decadal

timescale show a $R^2$ value of 0.50. We expected dynamic ocean processes such as the SPG strength to be controlling the $CO_2$ flux in the region, however the regression model show that the AMOCi and MLDi are predictors representing the dynamic ocean processes. The SST anomaly shows a negative anomaly on the interannual timescale and a positive anomaly on the decadal timescale, both correlating with a positive $CO_2$ flux anomaly (Fig. 6), which is also evident in the negative and postive correlation of the SSTi in the index analysis on interannual and decadal timescale, respectively (Table 3). This shows

that the relationship between the predictors might not be linearly, and that they in combination contribute to the $CO_2$ flux variability even though they individually do not contribute.

Lastly, the $CO_2$ flux variability in the SPE region is controlled by SST and MLD on both timescales, however, the interannual regression model also includes Wi, and the decadal regression model includes SSSi, as indicated in Fig. 7. The $R^2$ values of the regression models show that the correlation is better on the interannual timescale (0.55) than the decadal

timescale (0.40) (Table 4). Here, the regression model analysis show, that besides expected drivers such as SST and SSS (as discussed earlier and based on Eq.1-3), dynamical processes such as ocean mixing also drives the $CO_2$ flux variability. The MLD variability points to the SPE being a dynamical region, maybe affected by the SPG circulation in the western part of the Subpolar region. Furthermore, SPE is an ice-free region, allowing us to understand the dynamics of the $CO_2$ flux variability without the strong SIC driver. The decadal regression model for the SPE is the only one not including the Wi. The

wind speed anomaly is both negative and postive within the region, which indicates that the wind speed might not be a controlling factor in the SPE (Fig. 7).

**Table 4**
Applied regressions for all regions across timescales. $R^2$ values based on mean $R^2$ for all ten ensemble members in interannual (int) and decadal (dec) timescales

|  | NA | | NS | | NSE | | SPW | | SPE | |
| --- | --- | --- | --- | --- | --- | --- | --- | --- | --- | --- |
|  | Predictors | $R^2$ | Predictors | $R^2$ | Predictors | $R^2$ | Predictors | $R^2$ | Predictors | $R^2$ |
| Int | AMOCi, Wi | 0.48 | MLDi, SICi, Wi | 0.62 | MLDi, SSTi, Wi | 0.68 | AMOCi, SSTi, Wi | 0.68 | SSTi, MLDi, Wi | 0.55 |
| Dec | SPGi, SSSi, Wi | 0.37 | MLDi, SICi, Wi | 0.58 | NSGi, AMOCi, Wi | 0.53 | MLDi, SSTi, Wi | 0.50 | SSSi, MLDi, SSTi | 0.40 |



## 4. Discussion

### 4.1 Biases in ESM's

We calculated the $CO_2$ flux using the dependent parameters from the model as seen in Eq 1-3 to test to what degree the
model $CO_2$ flux could be reproduced. Using monthly mean fields of SST, SSS and wind speed we are able to reproduce the
$CO_2$ flux variability, however to reproduce the total flux we need to use a scaled wind field (wind x 1.2). PISCES calculates
the air-sea $CO_2$ flux daily (Döscher et al., 2021; Aumont et al., 2015), and by using monthly mean fields (as is the available
data) we risk a smoothening of the actual daily variability that can explain the need for a scaled wind field. Turner et al.
(1996) discusses how using monthly mean fields for wind and temperature can affect the calculation of atmopshere–ocean
fluxes (in their study dimethyl sulphide fluxes in the Nordic Seas, however using Wanninkhof (1992) gas transfer equtions
as in this study, Eq. 1-3). Reproducing the $CO_2$ flux help us understand the sensitivity and dependencies on the physical
parameters used to calculate the flux. The wind speed field shows a big sensitivity, which is also emphasised in the
regression models, all but one entailing the Wi.

    Earth System Models inevitably simplifies the complex climate system, which will introduce biases to the modelled
processes. One example being the monthly meaned wind field that introduces a bias in the calculated $CO_2$ flux. Another bias
adressed in this paper is the SIC bias in EC-Earth3-CC. As stated by Döscher et al., 2022; Tian et al., 2021 and shown in Fig.
1b EC-Earth3 (and thereby EC-Earth3-CC) shows an overestimation of the SIC in the Nordic Seas and the North Atlantic.
The index and regression model analysis of the NS shows that the SIC is a driver of the $CO_2$ flux variability, as expected
based on Eq. 3. Therefore, the overestimation of sea ice extent and concentration in EC-Earth3 will undoubtably affect and
possibly control the $CO_2$ flux variability in the ocean in the EC-Earth3-CC runs. However, even in regions of sea-ice such as
the SPW and the NA, SIC does not exclude signifcant impact of other processes, as SIC is not regarded as a predictor for the
$CO_2$ flux variability in these regions (Table 3, 4).

### 4.2 Predictors

Our analysis demonstrate that the $CO_2$ flux variability and its drivers cannot be considered across the larger northern North
Atlantic region, and demonstrate regional and sub-basin specifics. However, our analysis demonstrate that it is possible to
explain the $CO_2$ flux variability within the smaller regions to a good degree (table 4). Based on the regression models we
have established, that a linear correlation between the $CO_2$ flux and other predictors will not allow us to explain and
understand the full $CO_2$ flux variability. It is clear that internal dynamics between the different predictors also affect the $CO_2$
flux variability as shown in section 3.3. Furthermore, we also establish that not only the physical parameters SST, SSS and
wind speed work as drivers of the $CO_2$ flux in the regions. It is evident that wind speed poses a great control on the $CO_2$ flux
variability as discussed earlier, but we also see dynamic predictors such as ocean mixing (MLDi), gyre circulation
(NSGi/SPGi) and larger scale ocean circulation and transport (AMOC) as drivers. Holliday et al. (2020), Hátún et al. (2005,





2017) discuss how the larger scale dynamics such as the gyre circulation (SPG and NSG) and AMOC variability affect the

SST, SSS and PP in the northern North Atlantic, aligning with our findings. Despite being the smallesrt region in terms of area amog thise studies (Table 2), the NSE region exhibits the highest variability and mean flux. This remains the case until 1960 (Fig. 2), when the SPW and NSE flux timeseries converge. Our regression analysis yields the highest $R^2$ values in the NSE and SPW regions (Table 4), underscoring the robustness of our approach. Notably, the fact that our regression models provide the strongest explanatory power in the regions with the greatest variability and mean $CO_2$ flux suggests a stable and

reliable representation of the system dynamics.

Throughout our analysis we have established that a deepening of the MLD correlates with a positive $CO_2$ flux anomaly, explained by a deepening of the MLD resulting in an increase in DIC in the surface layer, decreasing $pCO_2$ in the surface layer and thus increasing the atmosphere-ocean $CO_2$ flux. We see this correlation both in the spatial patterns and in the index analysis in all the regions except for the SPE (Fig. 3-6, Table 3). Counter-intuitively, we see a (strong) negative

correlation between MLD and the $CO_2$ flux, both spatially and in the index analysis in the SPE region (Fig. 7, Table 3). Additionally, the MLDi is one of the predictors defining the regression models for both timescales in the SPE, which means that the MLD variability explains a big part of the $CO_2$ flux variability, despite their counter-intuitive negative correlation. The spatial patterns of the SPE shows a negative correlation between the $CO_2$ flux and both SSS and SST on both timescales (Fig. 7). Futhermore, the spatial patterns also show a possible blocking of the North Atlantic current, indicated by a closed

gyre system building up SSH. The blocking of the North Atlantic current might allow for fresh, cold and nutrient-rich Arctic water to enter the system, resulting in a positive $CO_2$ flux anomaly due to a decrease in temperature and nutrient supply increasing the PP and thus the $CO_2$ uptake. Increased freshwater supply from the Arctic would also result in increased stratification of the water coloumn, indicated by the shallowing of the MLD and thereby the negative correlation between the $CO_2$ flux and MLD. We have also found that in EC-Earth3-CC, PP and SSS is more or less replaceable, including spatial

patterns and their correlation. The North Atlantic is strongly influenced by the AMOC and SPG circulation, both regulating the salinity distribution and nutrient transport. The change in SSS reflects changes in the balance of freshwater input (i.e. freshwater from the Arctic). Both SSS and PP are sensitive to these large scale circulation dynamics, and they might co-vary, making them somewhat interchangeable in ESM's. These observations indicate that $CO_2$ flux variability in the northern North Atlantic cannot be adequately explained by individual predictors in a linear framework. Instead, a comprehensive

understanding of the underlying processes and the interactions among predictors is essential to accurately capture atmosphere-ocean $CO_2$ flux variability.



**4.3 Future projections**

The results of our regression model analysis allow us to consider whether we can predict the $CO_2$ flux variability in future
scenarios. Using the regression models for predicting the $CO_2$ flux variability in the North Atlantic on scenario runs from
EC-Earth3-CC or other ESM's would be beneficial for the understanding of the future climate system. We have defined the
regression model by using non-biogeochemical predictors, which makes it possible to apply them on data from ESM's that
do not contain a biogeochemical component such as EC-Earth3-CC. Looking into the near future, studies establish a

warming of the global climate system and the ocean (Fox-Kemper et al., 2021;  Bopp et al., 2013; Fu et al., 2016; Laufkötter
et al., 2016; Liu et al., 2023 Kwiatokowski et al., 2020). Ocean temperature increase would intuitively correlate with a
decreased atmosphere–ocean $CO_2$ flux, however, as demonstrated in this study the $CO_2$ flux variability is more complex than
simple linear relationships. An increased ocean temperature will result in a melting of the sea ice in the northern North
Atlantic, resulting in both an open ocean that can take up $CO_2$ , furthermore deepening the MLD by convection will also

increase the oceanic $CO_2$ uptake. However, a warming ocean will also affect the ocean circulation, both the large scale
circulation and the small scale ocean mixing. AMOC is predicted to slow down with global warming (McKinley et al., 2023;
Liu et al., 2023), stagnating the ocean circulation in the North Atlantic, and as seen in our analysis affecting the $CO_2$ flux
variability in particular the SPW region of the NA, but also further north in the NSE region. Furthermore, a warming ocean
and stagnated ocean circulation will result in increased stratification of the water column, trapping the cold $CO_2$ depleted

water masses in the deep ocean. The predicted global warming will possibly result in stronger winds in the North Atlantic
region (Lee et al., 2021, Ruosteenoja et al., 2019) which based on our results will inevitably increase the atmosphere–ocean
$CO_2$ flux. As discussed here, we see SIC and wind speed as main drivers of the $CO_2$ flux variability in the northern North
Atlantic, therefore an increase in ocean $CO_2$ uptake must be expected in the near future.

        Expanding the regression analysis to investigate their robustness in scenario runs will provide insight to whether

they can be used to actually predict the $CO_2$ flux variability in other ESM's that does not include any biogeochemical
processes. However, expanding this study to other model systems it is pivotal to have an in-depth understanding of the
model setup, i.e. with regard to the question of how gas transfer is handled across sea ice, and what other biases should be
considered. Our analysis provides a framework for identifying the key predictors and processes driving atmosphere-ocean
$CO_2$ flux variability in the northern North Atlantic. We demonstrate that physical ocean and atmospheric parameters alone

account for a significant portion of this variability, offering a valuable approach for assessing future $CO_2$ flux changes using
readily available climate model and reanalysis data.





**5. Conclusion**

Our analysis highlights the critical role of physical and dynamical processes in shaping $CO_2$ flux variability across the northern North Atlantic. Using historical simulations from the EC-Earth3-CC model, we show that key drivers, including SIC, SST, SSS, and wind stress, and ocean dynamics such as mixing and circulation, exert regionally and temporally varying influences on atmosphere–ocean $CO_2$ fluxes. While our regression models achieve high explanatory power, they capture interannual variability more effectively than decadal trends, underscoring the dominant role of short-term fluctuations.

Furthermore, our findings reveal that $CO_2$ flux variability cannot be attributed to simple linear relationships with individual predictors but instead emerges from complex interactions among multiple processes. Notably, wind stress exerts a particularly strong influence, aligning with expectations from gas transfer formulations. These results emphasize the spatially and temporally dependent nature of ocean carbon uptake and highlight the need for a multifaceted approach when assessing future $CO_2$ flux variability in a changing climate. Additionally, we conclude that the regression models we have defined in

our analysis can serve as a framework for predicting and understanding future atmosphere–ocean $CO_2$ fluxes in the ESM realm.

*Author contributions*. SMO and AP conceived and designed the study. AP wrote the majority of the manuscript, while SMO

provided supervision and guidance in analysing and interpreting the results. SMO and CRL contributed to the writing and revision of the manuscript.

*Code/data availability*. EC-Earth3-CC data is available through ESGF (*https://aims2.llnl.gov/search/cmip6/*) and postprocessing scripts are available upon request.


*Competing interest*. The authors declare that they have no known competing financial interests or personal relationships that could have appeared to influence the work reported in this paper. One of the authors, CRL, serves as an Associate Editor for Biogeosciences.

*Financial support*. AP was supported by SDU's climate cluster, EU Horizon Ocean observations and indicators for climate and assessments, ObsSea4Clim, Grant agreement ID: 101136548, 10.3030/101136548 Internal contribution Nr. 15, NCKF/DMI, the Villum Foundation (grant no. 29411) and by Danmarks Fri Forskningsfond (DFF, Carolin R. Löscher, grant no. 4283-00265B).




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
