# Peer review of "Northern North Atlantic climate variability controls on ocean carbon sinks in EC-Earth3-CC"

_EGUsphere, 2025_

## Referee Comment (RC2)

*Review of "Northern North Atlantic climate variability controls on ocean carbon sinks in EC-Earth-CC" by Pedersen et al.*

**Summary:**

The authors assess the role of dynamical and physical processes in the interannual-to-decadal variability of CO2 flux in the North Atlantic divided into four sub-regions, using the EC-Earth3-CC model. They start with a spatial composite analysis of CO2 fluxes and variables they identify as potential important drivers of the CO2 fluxes, including mixed layer depth (MLD), sea surface height (SSH), sea ice concentration (SIC), sea surface temperature (SST), sea surface salinity (SSS), air-sea pCO2 difference (dpCO2), and wind speed (W). This is followed by a correlation analysis with different indices. The authors find strong relations with several of the variables indicating complex dynamics, however, the wind speed stands out as the most important one.
Based on the results from the composite and correlation analysis, the authors construct linear regression models with which they can reconstruct up to 68 % of the modelled CO2 flux variability based on physical indicators only.
The authors address an important topic; understanding the variability of air-sea CO2 fluxes. The manuscript has the potential to be an important contribution to the community. However, before being published I think the authors need to work on the objective and the red line of the story, and potentially on their methods of analysis. More suggestions are found below.

**Major comments:**

- **Red line of the story:**
    1) The authors start off by "reconstructing" the simulated CO2 fluxes in their different regions of choice by using output of SST, SSS, W, dpCO2, SIC put into equations 1-3. The argue that this reconstruction "for our ability to explain from physical quantities the model variability." For this , I suppose that the SSS and SST goes into the K0 (the solubility). This gives the impression that the authors would like to reconstruct the simulated fluxes based on these 5 variables.
    2) Then, a composite analysis is performed. Apart from the five variables above (which goes into equation 1-3), the authors also include MLD, SSH.
    3) In the correlation analysis with different indices, additional variables(indices) are introduced.
    4) In the end, for the regression analysis, only a few of the indices in 3) are used.

Starting off as you do under 1), gives the impression that you will separate the contribution of each component in equation 1-3 to the variability. This is, in itself, an interesting question. But in the end, this is not what you do, which confuses the reader and make the story less coherent. Addressing this is a question of how you want to tell your story. Either you can do a local analysis and do a decomposition of each

component of these equations, which you also partly do in the composite analysis. When it comes to many of the other variables/indices, it is more about the relation to large scale dynamics/circulation (SSH, SPGi, AMOC), but that also impacts the local drivers (SSS, SST, dpCO2, SIC). If you want to keep all of these variables you could consider dividing the analysis part in to one local and large scale part. Otherwise, I would think that there is already enough material to only choose variables related to equations 1-3.

- **Relation to MLD:**

I think you need to consider removing the MLD from the analysis. Deeper mixed layers does not necessarily lead to increased CO2 uptake (locally). If the deepening of the mixed layer would bring water under (over) saturated in pCO2 to the surface, then it would allow for a local increase(decrease) in the CO2 uptake. However, in many cases it does not look like the pCO2 patterns perfectly match the MLD pattern in your composite analysis. In any case, if you would include dpCO2 in your regression model (see comment below), this effect would be taken into account.

Many of the aspects I mention above are also mentioned by the authors in the MS.

- **Relation to dpCO2:**

You have chosen indices that more or less covers all the parts of equation 1-3, except the dpCO2. If you want to base your study on equations 1-3, I would also include a dpCO2 index that you should try to put in the regression model.

At some point you mention that you want to reconstruct the CO2-fluxes based on physical indicators only. The reason for this is unclear, but in the discussion, you argue that with such a reconstruction model CO2 fluxes could be reconstructed also for Climate Models running without biogeochemistry. I find this argument not very strong since today the majority of ESMs are run with biogeochemistry, and we will unlikely go back to models without. Clarifying this would make the ***objective*** of the story clearer.

- **Relation to SSS and SST**

SSS and SST does not only impact K0, but also dpCO2. In your analysis of composite maps and correlation analysis of with SSTi och SSSi, do you know if the relation with the CO2 flux is mainly through the effect on pCO2 or on the K0?

- **Section 4.3:**

I think that you need to be very careful before you can project the results from your regression model based on interannual-to-decadal variability to climate change scenarios, which act at completely other time-scales. You already saw a big difference between your interannual and decadal time scales. I think that for this purpose ESMs run with biogeochemistry under future scenarios are more appropriate (you could of course consider doing a decomposition of drivers in future scenarios, but that is out of the scope of this paper I believe).

Something that I think you should emphasis more in the discussion is our ability to understand and predict interannual-to-decadal variations in ocean CO2 uptake. Some references here include:

Ilyina, T., Li, H., Spring, A., Müller, W. A., Bopp, L., Chikamoto, M. O., et al. (2021). Predictable variations of the carbon sinks and atmospheric $CO_2$ growth in a multi-model framework. *Geophysical Research Letters*, 48, e2020GL090695. **https://doi.org/10.1029/2020GL090695**

Li, H., Ilyina, T., Müller, W. *et al.* Decadal predictions of the North Atlantic $CO_2$ uptake. *Nat Commun* **7**, 11076 (2016). https://doi.org/10.1038/ncomms11076

Fransner, F., Counillon, F., Bethke, I., Tjiputra, J., Samuelsen, A., Nummelin, A., & Olsen, A. (2020). Ocean biogeochemical predictions—initialization and limits of predictability. *Frontiers in Marine Science*, 7, 386.

**Minor comments:**
L114: change maos to maps
L126: consider adding (CC) after carbon cycle
L128-129: atmospheric *chemical* composition ?
L130: The reference after NEMO3.6 should be Madec et al. , Döscher is for the full Earth System Model and should come after EC-Earth3 in the beginning of the sentence.
L132: Move the Aumont reference to just after PISCES-v2, and remove the Döscher reference, since it is not the reference for PISCES, but for EC-Earth
L133-136: It sounds from your description like the difference between EC-Earth3 and EC-Earth3-CC is that the CC one is run in an emission-driven mode. This would make sense because it would allow for an interactive carbon cycle and that carbon can be exchanged between the different model components. But in this case, since it was not run in an emission driven mode, what is the difference with EC-Earth3? It sounds like you in fact have been working with EC-Earth3 (which also include PISCES)?
L137: Here you come back to a description of PISCES, which you already started above. Please merge the two description parts.
L139: write  (P : N : C = 1 : 16 : 122) instead of (P / N / C, 1 / 16 / 122)
L139: if you are mentioning the ratios of P:N:C you should also mention how the other two nutrients are treated in the model. However, since you are not specifically looking into how the biological pump impacts the CO2 flux, you may remove this detail.
L146: U is the wind speed at 10 m above sea level?
L151-152:  I would not use the percentage sign since it is the sea ice fraction
L151-152:   you have not defined kgCO2 and k'gCO2
L154-162 and Figure 1: Please plot the gridded CO2 flux from the  Landschutzer data set next to your model CO2 flux, as well as the difference between the two. It will add much value to your paper to show the model performance compared to the observational-based dataset. Referring to figure 21b from Döscher does not add much value, since the reader needs to go to this paper and the map projection and colour map in that paper may be different.
L167: I do not see why you refer to figure 1b here?
L179: what kind of filter do you use for the filtering?
L208-210: "The level of reproducibility of simulated integrated fluxes from other parameters (Eq. 1-3) (F) and in particular their variability across time-scales from archived averaged monthly data will constitute an upper limit for our ability to explain

from physical quantities the model variability." This sentence is long and difficult to grasp. Consider reformulating and dividing in two. What does the (F) mean? Should it be after the simulated fluxes (F)?

Equations: The equations need numbering on the right-hand side.

L212: what do you mean by integrated flux? Should it be simulated flux?

L221-224: Specify that the SSS and SST dependency in this case is in the solubility constant (?). Otherwise, the reader may think about how these factors impacts pCO2 variability, which you are not investigating here.

Section 3.1: Consider changing the title to "Regional CO2 flux variability"

L248-253: You have provided explanations earlier for why most of these variables are expected to be important. However, you have not provided any arguments for SSH and MLD. Please provide a short line or two explaining how they can impact the CO2 flux.

Section3.1: when comparing the spatial pattern of the composites of co2 flux with other variables, you write "correlates", but you do not mention anywhere that you have done a spatial correlation of these composites. You may consider if such a correlation would be of value. If you choose not to do one, I suggest that you write "coincides" instead of "correlates".

L292-294: see major comment 2

L332-334: How is more saline water introduced to the system?

L357-356: see major comment 2

L530 Using monthly mean fields of SST, SSS and wind speed and pCO2 ?

L566-570: I do not see this positive  relation very clear your regions, especially between the pCO2 and the MLD. For the Subpolar East the negative relation is very clear on interannual time scales yes.

---

## Author Response (AR1)

Reply to R1:

We thank R1 for the positive review of our manuscript. We are pleased to find that R1 finds the work interesting and useful.
We find both the major and minor comments useful, and propose to incorporate them in the paper as described below:

L590-591: We acknowledge that this phrasing was unclear, and propose to rephrase to: "The results of our regression model analysis we feel confident that we can use the regression models defined in this study to predict the $CO_2$ flux variability in future EC-Earth3-CC scenario runs. Using the regression models for predicting the $CO_2$ flux variability in the North Atlantic on scenario runs from EC-Earth3-CC and potentially other ESMs would be beneficial for the understanding of the future climate system"

L591-592: We agree that using the regression models set up for EC-Earth3-CC on other ESMs might be complex and propose to include a discussion emphasising the possible intermodel differences and outline future perspectives with this in mind. A collaborative study across a number of ESMs comparing the primary predictors on a regional basis and their sensitivities (regression coefficients) could lead to an interesting synthesis and new insights.

L556-557: R1 rightly addresses the issue of correlated predictors which we address methodologically by limiting the number of selected predictors in the regression modeles by requiring a certain model improvement for each predictor. This could be explored further but we propose to add instead a cautionary note on the possible issues and our way around it which we find robust. For the MLD and winds it is true that they cannot be expected to be independent, but MLD is as a proxy for the upper ocean state and dynamics, whereas wind speed is directly influencing the $CO_2$ flux variability through the gas exchange equations.

L560-561: It definitely highlights the NSE region as an interesting region to study, and emphasises the need to investigate the dynamics of the smaller regions individually. Figure 2 shows the weighted mean $CO_2$ flux variability - the NSE is showing the highest variability and mean flux in relation to it's size, but the overall flux of the full North Atlantic is the greastest if you do not look at the weighted mean. The larger regions defined and regions influenced by sea ice in general show an interplay of processes where partly cancelating local anomalies also influence the average level of variability. A further stratification in sub regions is not considered constructive. We have balanced defining regions with (model dependent) dynamical characteristics and still geographically recognisable and partly established.

L569-586: Yes, we agree. We propose to include in the conclusion a statement on this finding along the lines suggested: "The main conclusion is that the $CO_2$ flux variability cannot be attributed to simple linear relationships with individual predictors but instead emerges from complex interactions among multiple processes."

We thank R1 for the rest of the minor comments, and if not commented above, they will be included in the revised manuscript.

Reply to R2:

We are pleased to read that R2 finds that 'the manuscript has the potential to be an important contribution to the community' and thank the reviewer for a thorough, constructive and generally positive review. We acknowledge the apparent need for a more clear motivation for our approach and scope of the paper as discussed by R2. By doing this it will also become clear why we have chosen the specific methods of analysis including working with indicators of ocean dynamics. We would like to note that this has not been raised as an issue with R1 which on the contrary nicely summarise and support our approach and storyline as follows:
"The novelty of the current work is the establishment of which predictors that can explain the CO2 flux variability in five regions in the North Atantic that are subject to quite different dynamic processes and atmospheric forcings."
"The concept of the work is interesting and useful, especially because it allows connecting basic variables and processes that can be obtained from different ocean climate and biogeochemical models commonly used to study climate change."
Still we see a need to better guide the readers as highlighted by R2. To address this we propose specifically to revise and clarify the introduction paragraph 1.1 (L104-122) clarifying the objective and logical progression of the sections.
Furthermore, other revisions suggested below will serve to address this concern. The replies will be listed in order of the review.

Relation to MLD:
We argue that MLD represents the process of vertical mixing, which is not necessarily represented directly by other indices. We have chosen to use MLD to be able to describe and include dynamical processes indirectly affecting the $CO_2$ flux variability such as ocean mixing. The different oceanic metrics and indicators will be partly correlated and interlinked through forcing and dynamics. See also comment to R1 on how we limit the predictors and regression models.

Relation to ΔpCO2:
Our focus on physical parameters is addressed above. Still R2 is correct that repeating the arguments for not including any non-physical parameters will be useful when it comes to the discussion on ΔpCO2 as an otherwise central and obvious parameter in describing the $CO_2$ flux variability. Rephrasing of section 1.1 L104-122 as proposed above is one step in clarifying this issue.

Section 4.3:
The authors agree with this point to some degree, and suggest to rephrase the discussion point to focus on the possibility of using the regression model to predict the $CO_2$ flux variability directly on scenario data from EC-Earth3-CC. However, we do believe that the regression models defined in this study could form a solid framework of explaining the $CO_2$ flux variability in future scenarios from other ESMs or even uncopuled ocean-only simulations. They might not be applicable directly, but could work as a starting point for explaining the future $CO_2$ flux variability and trends.
Both reviewers have commented on the discussions in this section and we will rephrase section 4.3 to modify the discussion with an emphasis on both R1's and R2's perspective. We propose to refocus the discussion towards using the regression models to predict the $CO_2$ flux in EC-Earth3-CC scenario runs, and to expand the discussion of using the regression models

on other ESMs. The authors agree that the regression models defined in this study will not be directly applicable to other ESMs, to be elaborated in the revised discussion section. Furthermore, we will include the relevant references kindly highlighted by R2 as a discussion point on the ability to understand and predict interannual-to-decadal variations in ocean $CO_2$ uptake.

We thank the reviewer for the minor comments and all of them will be incorporated in the revised manuscript as the authors believe they will improve the manuscript. Comments referring to major comment two has been addressed above. A few of the minor comments are commented below:

L130-152: We thank the reviewer for these clarifications and acknowledge that L133-136 was unclear. We propose to rephrase L133-136 to: "The configuration allows for simulations with emissions forcings, and the $CO_2$ flux is calculated from and proportional to the difference in partial pressure of ($\Delta pCO_2$) between the atmosphere and the surface of the ocean (Döscher et al., 2022)." as this is the accurate description of the dataset used. We thank the reviewer for noticing the mistake.

L154-162: We thank R2 for the suggestion and will update figure 1 with a subplot showing the gridded $CO_2$ flux from Landschutzer, so it is possible for the reader to visually compare the observational $CO_2$ flux with the EC-Earth3-CC $CO_2$ flux.

L208-210: This sentence will be rephrased based on R2's comments to clarify the meaning and to enhance the readability: "The reproducibility of simulated intregrated fluxes (F) derived from other parameters (Eq. 1-3), particularly their variability across different timescales, provides a useful benchmark. It sets an upper limit on how much of the model variability we can expect to explain using physical quantities from archived monthly-averages data. "

L221-224: The authors also agree with this point and will rephrase as follows: "It is also expected that the $CO_2$ flux variability is dependent on SST and SSS variability, however the effect of SST and SSS on the solubility constant ($K_0$) is too small to be considered important in these calculations, and the SST and SSS components of Eq 1-3 is therefore not scaled."

L248-253: We thank the reviewer for pointing this out and suggest to rephrase this section to: "These parameters include parameters already presented above (SST, SSS, SIC, $\Delta pCO_2$ and wind), however for the next part of the analysis we add mixed layer depth (MLD) and sea surface height (SSH). These parameters represent larger scale dynamics such as ocean circulation (SSH, gyre strength) and vertical mixing (MLD), which are candidates to be indirect processes controlling the $CO_2$ flux variability in EC-Earth3-CC."

Section 3.1: The authors agree with this clarification and has reworded the section using 'compares spatially' or 'mirroring' and not 'correlates'. Furthermore, we thank the reviewer for suggesting a new title for the section, which will be added in the revised manuscript.

L566-570: The authors agree that the description does not reflect the figures well, and have reworded the paragraph in the revised manuscript, focussing on the counter-intuitive patterns of the MLD.